# CADMA-Chem: A Computational Protocol Based on Chemical Properties Aimed to Design Multifunctional Antioxidants

**DOI:** 10.3390/ijms232113246

**Published:** 2022-10-31

**Authors:** Eduardo Gabriel Guzman-Lopez, Miguel Reina, Adriana Perez-Gonzalez, Misaela Francisco-Marquez, Luis Felipe Hernandez-Ayala, Romina Castañeda-Arriaga, Annia Galano

**Affiliations:** 1Departamento de Química, Universidad Autónoma Metropolitana-Iztapalapa, Av. Ferrocarril San Rafael Atlixco 186, Col. Leyes de Reforma 1A Sección, Mexico City 09310, Mexico; 2Departamento de Química Inorgánica y Nuclear, Facultad de Química, Universidad Nacional Autónoma de México, Mexico City 04510, Mexico; 3CONACYT-Universidad Autónoma Metropolitana-Iztapalapa, Av. Ferrocarril San Rafael Atlixco 186, Col. Leyes de Reforma 1A Sección, Mexico City 09310, Mexico; 4Instituto Politécnico Nacional-UPIICSA, Té 950, Col. Granjas México, Mexico City 08400, Mexico

**Keywords:** molecular design, ADME, toxicity, QSAR, selection score, reactivity indexes, kinetics, melatonin

## Abstract

A computational protocol aimed to design new antioxidants with versatile behavior is presented. It is called Computer-Assisted Design of Multifunctional Antioxidants and is based on chemical properties (CADMA-Chem). The desired multi-functionality consists of in different methods of antioxidant protection combined with neuroprotection, although the protocol can also be used to pursue other health benefits. The dM38 melatonin derivative is used as a study case to illustrate the protocol in detail. This was found to be a highly promising candidate for the treatment of neurodegeneration, in particular Parkinson’s and Alzheimer’s diseases. This also has the desired properties of an oral-drug, which is significantly better than Trolox for scavenging free radicals, and has chelates redox metals, prevents the ^●^OH production, via Fenton-like reactions, repairs oxidative damage in biomolecules (lipids, proteins, and DNA), and acts as a polygenic neuroprotector by inhibiting catechol-O-methyl transferase (COMT), acetylcholinesterase (AChE) and monoamine oxidase B (MAOB). To the best of our best knowledge, CADMA-Chem is currently the only protocol that simultaneously involves the analyses of drug-like behavior, toxicity, manufacturability, versatile antioxidant protection, and receptor–ligand binding affinities. It is expected to provide a starting point that helps to accelerate the discovery of oral drugs with the potential to prevent, or slow down, multifactorial human health disorders.

## 1. Introduction

The remarkable progress of medical sciences in the last century has led, for example, to a significant decline in infectious diseases worldwide [1]. This progress, together with an improvement in human habits (diet, exercise, non-smoking, etc.), has drastically increased people’s lifespans. According to the United Nations, the world population tripled from 1950 to 2020, while life expectancy at birth went from 64.2 years in 1990 to 72.6 in 2019 (and it is projected to reach 77.1 years in 2050) [2]. This means that the global population is getting older at an accelerated pace. However, health span has expanded in a much slower way. Thus, health at older ages is not only a current concern, but an urgent issue to be address [3,4]. Unfortunately, disease-free longevity is not likely in the foreseeable future. It has been claimed that it is time to focus on prolonging health and not only preventing death, i.e., “improving healthspan, not just lifespan” [1].

One of the factors contributing most to the lagging of life-quality, compared to its extension, is the myriad of chronic degenerative diseases that affects more than half of individuals over the age of 70 years old [4]. The main reason why there are still no efficient treatments for most of them is that they are multifactorial disorders. Some examples of these disorders, also known as polygenic, are: Cardiovascular diseases, cancer, metabolic, musculoskeletal, non-alcoholic fatty liver, and neurodegenerative diseases [5,6,7,8,9,10,11,12,13,14,15,16,17,18,19,20,21,22,23,24,25,26,27,28,29,30]. The latter are considered among the leading causes of death for elderly people [31,32]. It has been reported that the number of people affected by Parkinson’s and Alzheimer’s diseases is more than 10 and 6.5 million, respectively [31,32,33,34].

Neurodegeneration is characterized by the excessive loss of neurons, which is triggered by a wide variety of environmental, physiological, and genetic factors [31]. Oxidative stress (OS) has been identified as one of the main pathological factors promoting neuronal degradation [31,35,36,37,38,39,40,41,42,43,44,45,46,47,48,49,50,51,52,53,54,55,56,57]. Excessive exposure to reactive oxygen species (ROS) compromises the integrity of biomolecules (such as lipids, DNA and proteins), ultimately causing necrosis and cell death [58]. Since the human brain is rich in lipids and consumes large quantities of oxygen [59] (from which ROS are produced), it is highly susceptible to OS [60]. This chemical stress has been held responsible for the biomolecular alterations linked to several neurodegenerative diseases, including amyotrophic lateral sclerosis (ALS), Huntington’s (HD), Alzheimer’s (AD) and Parkinson’s (PD) diseases [31].

The DNA bases, particularly guanine, are easily oxidized by ROS. Such damage has been proposed to be the main cause of PD [61]. However, lipids are building blocks in neuronal membranes, which act as barriers and assure proper functioning [62]. They are highly susceptible of being attacked by ROS, or other free radicals, which produces lipid peroxidation. This process affects the membranes fluidity and permeability, and compromises the integrity of enzymes and receptors [63]. In addition, ROS production can be exacerbated by the presence of redox metal ions, such as copper or iron [64]. Thus, metal homeostasis is a key aspect regarding OS and neurodegeneration.

While there are currently no therapies that prevent or slow down PD and AD [65], some progress has been made to alleviate the symptoms. Acetylcholinesterase (AChE) inhibitors are used as symptomatic therapies for AD [26,66,67], while catechol-O-methyl transferase (COMT) [68,69,70,71] and monoamine oxidase B (MAOB) [72,73,74,75] inhibitors are used for PD. In addition, antioxidant-based therapies are emerging as promising complements since they improve neurocognitive performances and prevent excessive neuronal loss [50]. Polyphenols, in particular, have diverse neuroprotective effects. They have been reported to ameliorate cognitive impairment, increase brain plasticity, reduce brain edema and blood-brain barrier (BBB) leakage, restore lipid and metal homeostasis, and reduce mitochondrial disfunction and inflammation [28,51,57,76].

Multifunctionality has become the new paradigm in the design of drugs aimed to treat multifactorial diseases [77,78], including neurodegenerative disorders [79,80,81,82,83,84,85,86,87,88,89,90,91,92,93,94]. Some of the advantages of these drugs, over drug-cocktails or coformulations, are: (i) Simplified therapeutic regimes, (ii) reduced risks of drug interactions, (iii) less complex pharmacodynamics and pharmacokinetics, (iv) additive, or synergistic, therapeutic responses, and (v) no increased side effects [64,77,78,95]. In this context, it has been pointed out that computer-assisted approaches are a valuable support, which allows saving money, time and human efforts, as well as to reduce the number of experiments on animals [78].

Here, a computational protocol called CADMA-Chem (Computer-Assisted Design of Multifunctional Antioxidants, based on chemical properties) is presented. It was designed to identify viable candidates in the treatment of multifactorial diseases. Thus far, it has been used in the search of neuroprotection. However, it can also be useful in the pursuit of other health benefits. CADMA-Chem is the result of years of investigation and has the peculiarity of considering a wide diversity of criteria to select the most promising candidates. They are: Drug-like physicochemical profile, toxicity, manufacturability, free radical scavenging activity, metal chelation, capability of repairing oxidatively damaged biological molecules, and multi-target ligand behavior (specifically concerning AChE, COMT and MAOB).

Compared to other computational strategies, which are meant to design medical drugs, the major disadvantages of this protocol are that it is laborious and some of the investigated processes are complex to model. On the contrary, it has the following advantages: (a) It considers not only drug–receptor interactions, but also other relevant chemical features such as ADME properties, acid/base and tautomeric equilibria, and antioxidant vs. pro-oxidant behavior in the presence of redox metals; (b) it evaluates the potential toxicity of the candidates and their synthetic accessibility; (c) it involves only moderate structural modifications, thus the health benefits of the parent molecule are expected to be inherited by the new molecules. In other words, to the best of our knowledge, CADMA-Chem is the most complete computational protocol currently available for designing multifunctional medical drugs. It hopefully might contribute to the development of efficient treatments against neurodegenerative or other multifactorial diseases.

## 2. Results and Discussion

### 2.1. The CADMA-Chem Protocol

The idea was to develop a computational protocol for designing multifunctional antioxidants with the following desirable properties:Drug-like behavior (i.e., adequate permeation and bioavailability).Low toxicity.Easy manufacturability.Free radical scavenging capability.Metal chelation properties (^●^OH inactivating ligand behavior).Efficient for repairing oxidatively damaged biological targets (lipids, DNA and proteins).Polygenic neuroprotection, i.e., inhibitors of COMT, AChE and/or MAOB.

The hypothesis behind CADMA-Chem is two-fold: (i) A chemical with most of the previously mentioned properties should have neuroprotective effects, in particular against Parkinson’s and/or Alzheimer’s diseases (depending on the results from point 7). (ii) Derivatives with small structural modifications on a molecular framework should keep the benefits of the parent molecule (neuroprotection, for example).

Three main stages are involved in the protocol, namely, building the candidates, sampling the search space, and evaluating the candidate’s potential for the intended purpose. These are detailed in the following.

#### 2.1.1. Building the Candidates

To build the candidates, CADMA-Chem takes advantage of existing knowledge on the molecules previously used, to some extent, for the desirable behavior (for example, neuroprotection). Once the parent molecule is chosen, derivatives are built by moderate structural modifications, i.e., by adding up to three functional groups, such as: -OH, -NH_2_, -SH and -COOH). These groups are chosen based on their appealing properties:i.They can influence the acid-base behavior, thus modulating the proportion of neutral species at specific pH values, which is important for drugs passing across lipid barriers via passive diffusion.ii.They may contribute to increased free radical scavenging activity (via H or electron donation.iii.They may contribute to increased metal chelating capability.

#### 2.1.2. Sampling the Search Space

Schneider and Fechner [96] have called attention to several aspects that are important to consider when sampling the search space. They are:-To consider that an effective drug molecule is subject to more objectives than the binding affinity.-To define positive design restricts, which are those properties that allow for identifying the chemical subspace with a higher probability of containing drug-like molecules.-To define negative design restricts, or ‘tabu zones’, which are characterized by adverse properties and/or unwanted structures.-To reformulate the multi-objective problem into a single objective using a weighted score function. In such a function, the individual objectives are summed and frequently multiplied by a weighting factor.

The selection score (S^S^, Appendix A) is the scoring function in CADMA-Chem. The positive design restricts are the physicochemical parameters relevant for absorption, distribution, metabolism and excretion (ADME) properties. There are eight terms in S^S^ that allows for evaluating whether the candidates fulfill the Lipinski’s rule of five [97], the Ghose’s rule [98], and the Veber’s criteria (Table 1) [99]. For MW and logP, a unified criterion is used, thus Lipinski’s and Ghose’s rules are simultaneously fulfilled: 160 ≤ MW ≤ 480 and −0.4 ≤ logP ≤ 5.

The negative design restrictions are related to three other terms—two directly related to toxicity (Ames mutagenicity and the oral rat 50 percent lethal dose), and one for manufacturability (synthetic accessibility). S^S^ is calculated from them in such a way that the higher its value, the more likely the drug-like behavior of a molecule, the lower its toxicity and the easier its manufacturability. The value of this score, for each candidate, is compared to that of the parent molecules and to the average value for the reference set of the molecules (those in Appendix A for the study case presented here).

However, since the S^S^ includes eleven terms, a good value might mask particular failures. To avoid that, elimination scores (S^E^, Appendix A) were also used [100,101]. They allow for verification of whether any candidate deviates significantly from the average value of the reference set in any of its properties. However, it seems relevant to mention the importance of carefully checking what is causing high S^E^ values, since large deviations might arise from both undesired or desired behaviors. For example, there will be no reason to reject a candidate if its toxicity is much lower than the average of the reference set.

The first selection of potential candidates (subset 1) is made based on these scores, i.e., S^S^ and S^E^. Electronic structure calculations are performed, for subset 1, to estimate reactivity indexes. In turn, they are used to make the first assessment of the candidates’ likeliness for scavenging free radicals (SFR). The main chemical routes involved in SFR are single electron transfer (SET) and formal hydrogen atom transfer (*f*-HAT) reactions. Therefore, the most straightforward indexes for anticipating SFR processes are ionization energies (IE) and bond dissociation energies (BDE). They are the indexes used here to construct the electron and hydrogen donating ability map for antioxidants (eH-DAMA, Figure 1). Based on the candidate’s location on this map, the best is chosen (usually up to 6) to continue the investigation (subset 2).

Some comments on the eH-DAMA seems worthwhile:-This map is meant to analyze SFR processes for free radicals that are natural targets of antioxidants, for example peroxyl radicals, i.e., not highly reactive ones. If it is used otherwise, it might be misleading. A typical case would be a radical, such as ^●^OH, that usually reacts with antioxidants (via electron transfer) in a highly exergonic way. Such a reaction would be in the inverted region of the Marcus parabola. Consequently, albeit thermochemically viable, it may be a very slow reaction, not significantly contributing to antioxidant activity.-It is useful to include the target radical in the map (for example: ^●^OOH), as well as some reference antioxidants (for example: Trolox, ascorbic acid, and/or α-tocopherol).-The species located at the bottom-left of the map are those expected to be the best free radical scavengers, via SET and *f*-HAT.

#### 2.1.3. Evaluating Multifunctional Antioxidant Behavior

Chemical antioxidant activity (AOX) is a complex, and multifaceted, process that involves one or more of the following aspects:-Free radical scavenging activity (AOX-I) accounts for AOX activity in the absence of redox metal ions.-^●^OH inactivating ligand behavior (OIL, AOX-II) accounts for AOX in the presence of redox metal ions.-Repair of biological molecules (AOX-III).

Antioxidants can be specific or versatile, depending on their capability to offer protection by one or more of the above-mentioned processes. In addition, they may present acid-base equilibria, which affects both reactivity and membrane permeability. Thus, all these aspects should be studied in detail when looking for multifunctional antioxidants.

##### 2.1.3.1. Molar Fractions at Physiological pH

The acid constants (*p*K*a*) of the candidates can be calculated with the fitted parameters approach (FPA), which is fast, easy to use, and reliable [96,102,103]. This involves using the equation: *p*K*a* = *m* ΔG_BA_ + C_0_, where ΔG_BA_ is the Gibbs energy difference between the conjugated base and the acid. The parameters (*m* and *C*_0_, i.e., the slope and intercept of the linear fit) are currently available, at numerous levels of theory, for phenols, amines, carboxylic acids, and thiols. Those used in this work are reported in Table 2.

After knowing the *p*K*a*s of each candidate, their deprotonation routes are elucidated, and the molar fraction of the acid base species are estimated. Molecules with a negligible neutral fraction (lower than 1%) are not considered further. The reason is that multifunctional antioxidants, with oral drug like behavior, are expected to enter the cells by passively crossing biological membranes.

##### 2.1.3.2. Free Radical Scavenging (AOX-I)

The investigation of AOX-I (for subset 2) includes thermochemistry, kinetics, and all possible reaction mechanisms and pathways. It is recommended to use Gibbs free energies for thermochemical analyses; thus, entropy is considered. For kinetics, there are some important aspects to keep in mind: (i) To include all the reaction sites when calculating the overall rate coefficients; (ii) to consider environmental conditions such as its polarity and *p*H; (iii) to include tunneling corrections for *f*-HAT reactions; (iv) to consider the reaction path degeneracy; (v) to consider the diffusion rate; and (vi) to use not very reactive radicals, such as the antioxidants counterpart, peroxyl radicals, are ideal for this role. These aspects are in line with the QM-ORSA protocol, which has been proven to produce rate constants that are in very good agreement with the experiments [104].

##### 2.1.3.3. OIL Behavior (AOX-II)

The ^●^OH-inactivating ligand (OIL) [105,106] behavior, or antioxidant activity type II (AOX-II), is also explored for subset 2. It can occur by lessening the reduction of metal ions (OIL-1) or by scavenging ^●^OH, just after they are produced via Fenton-like reactions (OIL-2) [107]. Metal chelation is involved in both. It can take place by, at least, two different mechanisms: Direct chelation (DCM) and coupled deprotonation-chelation (CDCM). Thermochemistry and/or kinetics are estimated for the different reaction path. Copper is used here as redox metal because of his known role in neurodegeneration [108,109,110,111,112,113]. The best protectors, against metal-induced oxidation, are identified based on these data.

##### 2.1.3.4. Repairing Biological Molecules (AOX-III)

The antioxidant activity type III for subset 2 is examined for three kinds of biomolecules: Lipids, proteins and DNA. The models used to represent each (Figure 1) are:-Lipids: A simplified model of linoleic acid (LM), with 2 allylic H atoms (the key chemical feature of easily oxidizable lipids), is used to represent unsaturated fatty acids [114].-Amino acid residues in proteins: Six residues, highly susceptible to OS [115,116,117,118,119,120,121], are considered, namely cysteine, histidine, leucine, methionine, tryptophan, and tyrosine. To represent them, the model known as the realistic model is used [120,122,123,124,125,126,127,128,129,130,131,132].-DNA: 2′-deoxyguanosine (2dG) was chosen for modeling DNA based on the fact that it is the most easily oxidized nucleoside [133,134,135,136,137].

**Scheme 1 ijms-23-13246-sch001:**
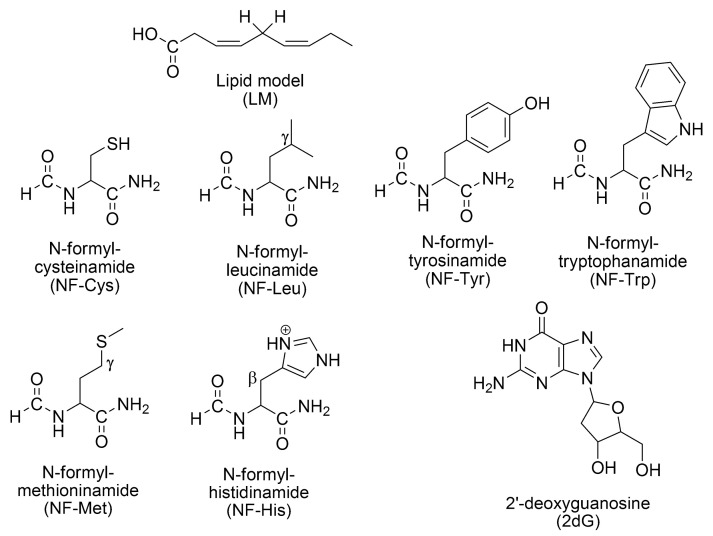
Models used to mimic biomolecules.

The main mechanisms involved in the oxidation on these biomolecules are:-Lipids: *f*-HAT, involving the allylic hydrogens.-Amino acid residues in proteins: SET from Tyr and Trp. *f*-HAT from Cys, Tyr, Leu, Met, and His.-DNA: SET from 2dG sites (the nucleoside most easily oxidizable) [138], *f*-HAT from the deoxyribose unit (yielding C-centered radicals) [139,140,141,142], RAF yielding the 8-OH-dG adduct, the precursor of one of the most abundant lesions in DNA: 8-oxo-7,8-dihydro-2′-deoxyguanosine (8-oxo-dG) [143], which is a biomarker of OS [144,145].

The corresponding chemical routes are explored as viable paths to repair oxidized DNA:a.Repairing guanine-centered radical cations, via SET.b.Repairing C-centered radicals, in the deoxyribose unit, via *f*-HAT.c.Repairing 8-OH-dG lesions via sequential hydrogen atom transfer followed by dehydration (SHATD) [146].

#### 2.1.4. Evaluating Polygenic Neuroprotection

The enzymatic interactions of molecules in subset 2 are evaluated using molecular docking. The interactions of the candidates with the catechol-O-methyltransferase (COMT), acetylcholinesterase (AChE) and monoamine oxidase B (MAOB) are explored. Binding energies are compared to those of known inhibitors and natural substrates. These enzymes were chosen based on their role in the etiology and treatment of neurodegenerative disorders, in particular Parkinson’s and Alzheimer’s diseases.

### 2.2. Study Case

More than 9000 candidates have been built, so far, using the CADMA-Chem protocol. They are derived from molecules with some neuroprotective effects (those contained in the reference set reported in Appendix A). Several of them have been investigated to some extent [147,148,149,150,151,152]. Here, the melatonin derivative, originally labeled as dM38 [153], is used to illustrate the details of the analyses.

#### 2.2.1. Building the Candidates

Using the CADMA-Chem strategy, 116 melatonin derivatives were previously built [153], considering four substitution sites (Figure 2). Five were identified as the most promising candidates [153]. One of them is dM38, which has been chosen here to explore multifunctional behavior.

#### 2.2.2. Sampling the Search Space

The physicochemical parameters of dM38, its parent molecule (melatonin), and the average for the reference set of molecules (^A^RS) are reported in Appendix A. As these values show, dM38 fulfill all the requirements in the Lipinski’s [97] and Ghose’s rules [98], as well as the Veber’s criteria [99], with only one exception. Its number of non-hydrogen atoms is 19. Thus, it is outside the 20 to 70 range recommended in the Ghose’s rule. However, this value does not suggest permeation or absorption issues. It is only one atom below the limit, and the number of non-hydrogen atoms in melatonin (which has no such issues) is even lower (17). The dM38 selection score (3.61, Appendix A) was found to be slighter lower than that of its parent molecule (3.75), and significantly above that of the reference set of molecules (3.0).

The elimination scores are also reported in this table and plotted in Figure 2 and Figure 3. The large values of S^E^ for melatonin in SA and M arises from its ease of synthesis and low mutagenicity. Thus, they are desirable deviations from the references. For dM38, the largest deviations correspond to the physicochemical properties. However, as explained above, it is expected to have a drug-like behavior. Its overall elimination score (S^E,ADMETSA^) was found to be lower than that of melatonin. Thus, in general, the investigated derivative has properties that are in line with the reference set of molecules already in use as neuroprotectors.

The eH-DAMA for dM38 is shown in Figure 4, while the reactivity indexes of each species (IE, EA and BDE) are reported in Appendix A. This map shows that melatonin would not be a good peroxyl radical scavenger, which is in line with previous reports [154]. On the contrary, its derivative dM38 is predicted to be excellent for such a purpose. In its mono-anionic form, which is the dominant one at physiological *p*H, it is expected to be more efficient than all the explored antioxidant references (i.e., Trolox, ascorbic acid and α-tocopherol) through both SET and *f*-HAT.

#### 2.2.3. Evaluating Multifunctional Antioxidant Behavior

##### 2.2.3.1. Molar Fractions at Physiological pH

The *p*K*a* values of dM38 were estimated to be 5.90 and 12.12 [153]. They correspond to the deprotonation route shown in Figure 3. The molar fraction estimated using these *p*K*a* values are 0.031 and 0.969 for the neutral and mono-anionic species (at pH = 7.4), respectively. The di-anion fraction is negligible (lower than 10^−4^). Accordingly, the analyses in Section 2.2.3.4 will be focused on the mono-anion in an aqueous environment, unless otherwise specified, since it represents the majority of the global dM38 population (96.9%) in such a media. It is important to note, however, that the neutral fraction is not negligible (3.1%) in this case, which guarantees passive crossing through biological membranes. On the contrary, in lipid media, where deprotonation is not viable, the neutral species prevails.

##### 2.2.3.2. Free Radical Scavenging (AOX-I)

The ^●^OOH scavenging activity of dM38 showed that the main reaction mechanism depends on solvent and *p*H [147]. In lipid media, where the neutral species is the predominant one, the *f*-HAT from the thiol group accounts for 99.5% of the total reaction. On the contrary, in aqueous solution, the thiolate anion is responsible for the SFR process, which involves 74.6% SET and 25.4% *f*-HAT of the phenol group.

The overall rate coefficients were estimated to be 9.96 × 10^9^ and 5.17 × 10^6^ M^−1^ s^−1^ in water and lipid media, respectively. This means that the free radical scavenging activity of dM38 exceeds those of Trolox, ascorbic acid and α-tocopherol in both environments. The rate coefficients for the reactions between these reference antioxidants and ^●^OOH were reported to be (in non-polar media) 3.40 × 10^3^ [155], 5.71 × 10^3^ [104], and 3 × 10^6^ M^−1^ s^−1^ [156], Ingold, respectively. In aqueous environments, however, they are 8.96 × 10^4^ [155], 3.07 × 10^5^ [104], and 2 × 10^5^ M^−1^ s^−1^ [157], Bielski.

Thus, it can be said that, in lipid media (non-polar), dM38 is expected to scavenge peroxyl radicals 1520, 905 and 2 times faster than Trolox, ascorbic acid and α-tocopherol, respectively. In water solution (considering that the molar fraction of ^●^OOH, at *p*H = 7.4, is 0.0025), the dM38 efficiency as peroxyl scavenger surpasses those of the reference antioxidants 278, 81, and at least 125 times, respectively. It should be noticed that the last number is a lower limit since the reported value was measured in water/ethanol mixture (0.15/0.85) and at acid *p*H. Under such conditions, the rate constant is expected to be higher than that in water at physiological *p*H.

##### 2.2.3.3. OIL Behavior (AOX-II)

The calculations for the direct-chelation mechanism (DCM) are rather straightforward. However, those involving the coupled deprotonation–chelation mechanism (CDCM) requires taking the *p*H into account. This means that conditional Gibbs free energies of the reaction should be calculated at the *p*H of interest because, as the *p*H increases, so does the viability of the reactions involving deprotonation. Details on how to do so can be found elsewhere [158]. Here, physiological *p*H (taken as 7.4) is considered.

Different Cu(II) complexes were located (Figure 4). To calculated the Gibbs free energies of reaction, ‘free’ copper ions were modeled and coordinated for four water molecules and a square-planar-like arrangement [159]. For consistency, four water molecules were also included in the Cu(I) surroundings (two coordinated to copper and two solvating the system). The reactions were all found to be significantly exergonic (Appendix A). However, the CDCM-cSO complex (Figure 5) is, by far, the most abundant one. It accounts for 99.993% of the total complex population, according to the Maxwell–Boltzmann distribution.

The possible OIL behavior was analyzed for this complex, in particular for OIL-1, since lessening the reduction of metal ions is expected to prevent the first step of the Haber-Weiss reaction (HBR), and consequently, the ^•^OH production catalyzed by copper. For that purpose, two reductants were considered:-The ascorbate anion: A moderate reductant, which is frequently used in experiments to induce oxidative conditions (mixed with copper).-The superoxide radical anion (O_2_^•−^): A very strong reductant, present in biological systems and involved in Fenton-like reactions.

The obtained results (Table 3) show that the Cu(II) reduction by ascorbate is fully inhibited when the CDCM-cSO complex is formed. If the reductant is O_2_^•−^, the Cu(I) yield is expected to be dramatically lowered, although not completely turned off. However, the reaction would be more than 10^4^ times slower than that of ‘free’ Cu(II). These findings indicate that CDCM-cSO should be a very efficient OIL-1 that is capable of preventing the metal catalyzed ^•^OH production.

##### 2.2.3.4. Repairing Biological Damaged Molecules (AOX-III)

The lipid repairing process was modeled using the neutral form of dM38, since it is expected to take place in a non-polar aprotic environment. The *f*-HAT repairing of amino acid residues and 2dG sites in DNA, on the contrary, were modeled using the dM38 mono-anion, in water. The Gibbs free energies of the *f*-HAT reactions between dM38 and damaged biomolecules are reported in Table 4. It was found that the most likely H donor sites are the -SH group for the lipid repairing process and the phenolic -OH for the rest. Thus, kinetic calculations were performed for these *f*-HAT paths and for the SET reactions when thermochemically viable (Table 5 and Table 6).

While the reaction between LM and dM38 is slightly exergonic (Table 4) and its kinetics rather slow (Table 5), it is expected that this compound could repair oxidative damage in the bis-allylic site of lipids by *f*-HAT from the thiol site. For the amino acid residues, cysteine and tyrosine are expected to be the ones that dM38 repairs the fastest via *f*-HAT. The SET processes for tyrosine and tryptophane were also found to be very efficient for the neutral form of dM38. When the mono-anion is involved, the SET reactions significantly slow down since they are located in the inverted zone of the Marcus parabola due to their high exergonicity. The total SET rate constant (*k_tot, SET_*) was calculated considering the populations of the acid-base species, at *p*H = 7.4, as:*k_tot, SET_* = *^M^f* (dM38_neutral_) k(dM38_neutral_) + *^M^f* (dM38_anion_) k(dM38_anion_)(1)

Among the investigated amino acid residues, tyrosine is the only one that is expected to be repaired through both processes (*f*-HAT and SET). The damaged species in the first case is the tyrosine phenoxyl radical, and in the second case, the tyrosine radical cation. The corresponding lesions are predicted to be quickly repaired by dM38 since the associated rate constants are 1.74 × 10^7^ (Table 5) and 2.44 × 10^8^ M^−1^ s^−1^ (Table 6), respectively.

The repair of the three most common lesions in oxidatively damaged DNA were investigated. The corresponding chemical routes are:-Route I: The repair of 2dG-centered radical cations, via SET (Figure 5).-Route II: The repair of 2dG C4-centered radical, in the deoxyribose unit, via *f*-HAT (Figure 6).-Route III: The repair of 8-OH-dG lesions in 2dG, via SHATD (Figure 7).

**Scheme 5 ijms-23-13246-sch005:**
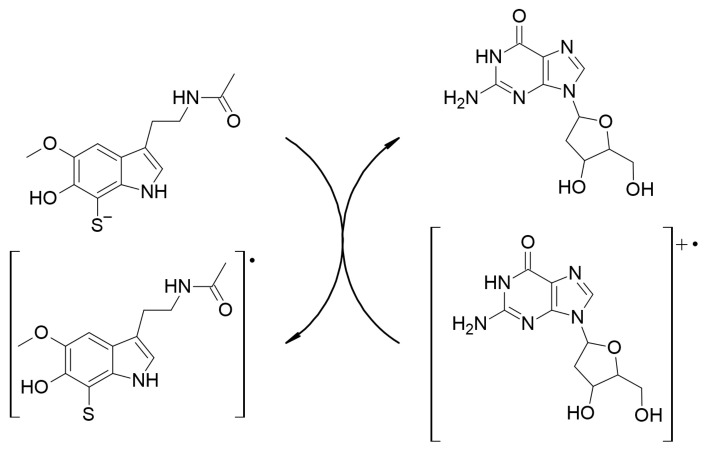
Repair of 2dG-centered radical cations, via SET, by dM38.

**Scheme 6 ijms-23-13246-sch006:**
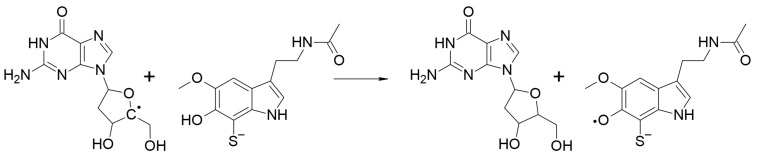
Repair of 2dG C4-centered radical, in the deoxyribose unit, via *f*-HAT, by dM38.

**Scheme 7 ijms-23-13246-sch007:**
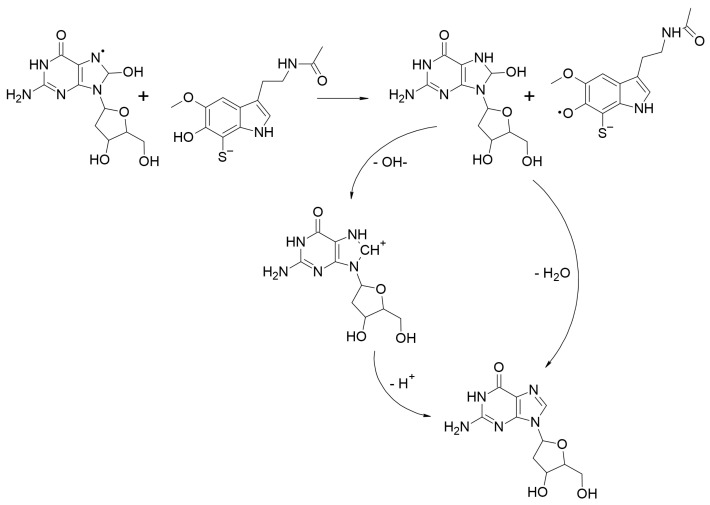
Repair of 8-OH-dG lesions in 2dG, via sequential hydrogen atom transfer followed by dehydration (SHATD), by dM38.

The obtained results indicate that the melatonin derivative dM38 is capable of reverting DNA damaged induced by oxidative stress. It was found to be particularly efficient at repairing 2dG-centered radical cations, via SET (route I) and 8-OH-dG lesions in 2dG, via SHATD (route III). The estimated rate constants for these processes are 2.45 × 10^8^ and 1.03 × 10^6^ M^−1^ s^−1^, respectively (Table 5 and Table 6). The 2dG C4-centered radical, in the deoxyribose unit, is also expected to be repaired, via *f*-HAT, at a slower rate.

The results obtained for dM38, regarding the repair of oxidatively damaged biomolecules, are very promising. They strongly suggest that this molecule may be capable of restoring lipids, proteins and DNA to their pristine forms. This kind of antioxidant protection would prevent, at least to some extent, permanent lesions and associated health disorders.

#### 2.2.4. Evaluating Polygenic Neuroprotection

The best docked poses of the investigated ligand–receptor complexes are provided in Figure 6. In all of them, the ligand is dM38. The receptors are the COMT, MAOB and AChE enzymes, which are known to be involved in neurodegeneration (especially in Parkinson’s and Alzheimer diseases).

COMT catalytic site comprises a Mg^2+^ cofactor, as well as the residues D141, K144, D169, N170, E199 and the S-adenosylmethionine fragment [160]. In the [dM38-COMT] complex (Figure 6A), the adduct is stabilized by the formation of several interactions. One of them with the metallic cofactor and the rest with four of the key residues. Surprisingly, the hard ion Mg^2+^ is bounded to the soft sulfide moiety. This unusual Mg-S bond (d = 2.07 Å) is possible because the deprotonated sulfur can form several H-bonds with the surrounded amino acids. Six H-bonds were found and involved residues M40, D141, K144, E199, N170, and M201; as well as some alkyl and π-interactions. However, a steric repulsion between an alkyl group in dM38 and tryptophan 38 (Y38) decreases the ΔG_U_ value by approximately 0.6 kcal/mol. The complete interaction path in the [dM38-COMT] complex is shown in Appendix A.

The AChE active site can be divided in six regions (catalytic, peripheral, acyl, anionic, oxyanionic and aromatic sites) and is formed by 18 residues [161]. In the [dM38-AChE] complex, the ligand connects with seven key residues (Figure 6B). The sulfide moiety is bounded to the tryptophan 86 (W86) by the π-S connection in the acyl region. Histidine 447 and serine 203 (H447, S203), in the catalytic site, are bounded through a π-π t-shaped interaction and an H-bond, respectively, to the aromatic ring, hydroxyl and methoxy groups in dM38. Tryptophan 341 (Y341) in the aromatic site participates in π-type interactions with the aromatic ring and acetamide fragments. Glycine 122 (G122) forms a H-bond with the methoxy group in the oxyanionic zone. In the anionic site, phenylalanine residues (F295 and F297) are bonded to the pyrrole, methyl and carboxyl groups forming π-alkyl and hydrogen interactions, respectively. It has been previously reported that compounds capable of binding to the anionic site could decrease the accumulation of β-amyloid peptide [162]. Therefore, the binding mode of dM38 suggests that this compound may have therapeutic potential against Alzheimer’s disease.

MAOB is a FAD-containing (flavin moiety) enzyme with a hydrophobic active site separated in two cavities. In the limit of these compartments, isoleucine 198 (I198) has the function of gatekeeper. Substrates and potential MAOB inhibitors should pass I198 and covalently bound to the FAD N5 atom. They are usually located in the region known as the aromatic cage formed by the flavin moiety and tyrosine residues Y398 and Y435. In the best docked pose of [dM38-MAOB] (Figure 6C), I198 forms an H-bond with the sulfur atom and a π-alkyl interaction with the aromatic part of the melatonin framework. Y398 interacts, through π-stacking, with the same aromatic fragment. Y435 interacts through π-σ with the methoxy group in dM38. Other H-bonds, involving cysteine 172 (C172) and glutamine 206 (Q206), minimize repulsion in the complex. An H-bond with the FAD cofactor completes the interaction path. The acetamide fragment in dM38 and the N5 atom are closely located, at 1.90 Å. This observation could be explained by a dipole produced by the electrophilic carboxyl group and the nucleophilic amine moiety. This conformation opens the possibility of covalent dM38-FAD binding. However, this hypothesis must be validated in future works using more accurate computational techniques.

The binding energies (G_U_) and inhibition constants (Ki) for the investigated ligand-receptor complexes are reported in Table 7. The ligand of interest is the melatonin derivative dM38. Its parent molecule, as well as the natural substrates of these enzymes, and known inhibitors, are included for comparison purposes.

The values estimated for inhibitors (i.e., tolcapone, safinamide and donepezil) are in good agreement with previous literature reports [160,163,164], which validates the reliability of the used docking protocol. The interactions of dM38 with the analyzed enzymes were found to be stronger than those of the corresponding natural substrates (i.e., higher G_U_ values, and lower Ki values). This finding suggests that this derivative may be an efficient inhibitor of COMT, AChE and MAOB, thus helping to prevent the degradation of dopamine, acetylcholine, and phenylethylamine.

It seems worthwhile to comment on the fact that the inhibitory efficiency of dM38 is predicted to be lower than those of the reference inhibitors. However, while they are specific inhibitors (only one target), dM38 is expected to act as a polygenic neuroprotector. It is also interesting to compare this derivative with its parent molecule, since the later has been reported to have neuroprotective effects [165,166,167,168]. For the three investigated enzymes, dM38 is predicted to be a better inhibitor than melatonin. The polygenic score (S^P^, Appendix A), shown in Figure 7, clearly shows the trend discussed above.

#### 2.2.5. CADMA-Chem Flowchart

A schematic representation of the sequential steps involved in this protocol is presented in Figure 8.

## 3. Materials and Methods

The software used, its details, and the properties calculated with them are reported in Table 8. More details on the procedures for molecular docking are reported in Appendix A. The expressions used to calculate the selection, elimination, and polygenic scores are provided in Appendix A.

## 4. Conclusions

CADMA-Chem is a protocol aimed to design multifunctional antioxidants. Here it has been applied to a melatonin derivative (dM38) to illustrate the whole procedure in detail. The multi-functionality searched for consisted of different ways that antioxidant activity combined with neuroprotection. However, the protocol can be used to design molecules with other health benefits, such as anticancer properties.

CADMA-Chem takes advantage of pharmaceutical drugs that have been already used for the desired purpose with some success. It is expected to provide a starting point that helps to accelerate the discovery of novel oral drugs with the potential to be used for ameliorating multifactorial health disorders, at low costs. To the best of our knowledge, CADMA-Chem is, currently, the only protocol that simultaneously involves the analyses of drug-like behavior, toxicity, manufacturability, versatile antioxidant protection, and receptor–ligand binding affinities.

The study case, i.e., dM38, seems to be a highly promising candidate for the treatment of neurodegeneration, in particular for Parkinson’s and Alzheimer’s diseases. It was found to have the desired properties of an oral-drug, to be significantly better than Trolox for scavenging free radicals, to chelated redox metals, preventing the ^●^OH production via Fenton-like reactions, to repair oxidative damage in biomolecules (lipids, proteins, and DNA), and to act as a polygenic neuroprotector by inhibiting COMT, AChE and MAOB.

It is hoped that the results presented here would promote further investigation into the subject, including the synthesis and experimental exploration of dM38 as a multifunctional antioxidant with neuroprotective properties.

## Data Availability

Data is contained within the article or Appendix A.

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
