# Peer review of "CADMA-Chem: A Computational Protocol Based on Chemical Properties Aimed to Design Multifunctional Antioxidants"

_ijms, 2022, doi:10.3390/ijms232113246_

Round 1

Reviewer 1 Report

OBSERVATIONS

 regarding the manuscript

CADMA-Chem: A Computational Protocol Based on Chemical 2 Properties Aimed to Design Multifunctional Antioxidants

In the paper CADMA-Chem: A Computational Protocol Based on Chemical 2 Properties Aimed to Design Multifunctional Antioxidants, the authors investigated the design of novel antioxidants with versatile behavior, using a computational protocol based on chemical properties, CADMA-Chem. Specifically, to illustrate the protocol in detail, the melatonin derivative dM38 was used. Its properties were presented, by comparison with Trolox. The application of CADMA-Chem in the simultaneous investigation of properties such as toxicity, manufacturability, versatile antioxidant protection and receptor-ligand binding affinities was pointed out. The paper is well documented and correctly structured, and the protocols used are adequate for the purpose of the study.

After reading the manuscript, I formulated the following observations:

1.     Pag. 6 - The authors say that it is useful to include in the map some reference antioxidants - Trolox, ascorbic acid and/or alpha-tocopherol, but the discussion is done only by comparison with Trolox. A comment regarding the other 2 specified antioxidants would be useful.

2.     Pag.7, 217 - Please specify the full name of the parameters m and C0

3.     Pag.9, R 290 - Please provide bibliographic references for the most 5 promising candidates, one of them beeing dM38

4.     Pag. 13, Table 5 - Please add ΔG and its measurement unit to the table.

5.     Pag. 13, Table 6 - Please add the unit of measure of the constant k.

6.     Pag. 19, Fig. 7 - Please add Sp on the Oy axis.

I noticed that the Author Galano A. has around 18 citations among the bibliographic references. The number seems high to me, but it is up to the editor if he accepts this aspect, considering that the cited works are related to the subject of the work.

Author Response

1. pag 6 - The authors say that it is useful to include in the map some reference antioxidants - Trolox, ascorbic acid and/or alpha-tocopherol, but the discussion is done only by comparison with Trolox. A comment regarding the other 2 specified antioxidants would be useful.

RESPONSE: The comparisons with reference antioxidants now includes the three of them (Trolox, ascorbic acid and alpha-tocopherol).

2. pag 7, 217 - Please specify the full name of the parameters m and C0

RESPONSE: The correction is done.

3. pag 9, R 290 - Please provide bibliographic references for the most 5 promising candidates, one of them beeing dM38

RESPONSE: The reference was included, as suggested.

4. pag 13, Table 5 - Please add ΔG and its measurement unit to the table.

RESPONSE: The correction is done.

5. pag 13, Table 6 - Please add the unit of measure of the constant k.

RESPONSE: The units of the rate constant (k) are included in the corresponding column head.

6. pag 19, Fig. 7 - Please add Sp on the Oy axis.

RESPONSE: Figure 7 was corrected as suggested.

I noticed that the Author Galano A. has around 18 citations among the bibliographic references. The number seems high to me, but it is up to the editor if he accepts this aspect, considering that the cited works are related to the subject of the work.

RESPONSE: They are, in fact, directly related to the subject of the work. In addition, this manuscript includes almost 180 references.

Reviewer 2 Report

The manuscript introduced a multifunctional antioxidant design method by computational protocol which was interesting. Comments were listed below:

1 avoid abbreviation during their first appearance.

2 In the introduction part, the background of drug design was introduced but the part for CADMA-chem (its merits and shortage) was insufficient.

3 The possible synthetic path of dm38 should be mentioned.

4 Maybe some experimental performance of dm38 as antioxidant can be involved.

Author Response

1 avoid abbreviation during their first appearance.

RESPONSE: The manuscript has been corrected accordingly.

2 In the introduction part, the background of drug design was introduced but the part for CADMA-chem (its merits and shortage) was insufficient.

RESPONSE: A paragraph has been added at the end of the Introduction section. The advantages and disadvantages of CADMA-Chem, compared to other computational protocols aimed to design medical drugs, are discussed.

3 The possible synthetic path of dm38 should be mentioned.

RESPONSE: This derivative (dm38) has not been synthesized yet. Actually, it is proposed as a promising candidate to be synthesized in this manuscript. This aspect exceeds the goals of the presented protocol. However, we hope that the reported properties of this molecule will attract the interest of experimentalist and it will be synthesized soon.

4 Maybe some experimental performance of dm38 as antioxidant can be involved.

RESPONSE: This molecule has not been synthesized yet. Thus, there is still no experimental evidence on its antioxidant properties.

Reviewer 3 Report

The topic of this work is interesting by introducing a protocol of research. The scientific content is appropriate. But the architecture of this manuscript is not easy to understand what the authors did exactly. I strongly suggest the authors to prepare a schematic representation of their work in a flowchart or something like it to show steps of the work, data entry, concepts, algorithms, analysis, data output, and etc.

In the current version, the work is not easy to understand.

Author Response

The topic of this work is interesting by introducing a protocol of research. The scientific content is appropriate. But the architecture of this manuscript is not easy to understand what the authors did exactly. I strongly suggest the authors to prepare a schematic representation of their work in a flowchart or something like it to show steps of the work, data entry, concepts, algorithms, analysis, data output, and etc. In the current version, the work is not easy to understand.

RESPONSE: The suggested flowchart has been included in the new version of the manuscript (section 4.5, figure 8).

Round 2

Reviewer 3 Report

The current revised version is suitable for publication.